# Low-Cost Zinc–Alginate-Based Hydrogel–Polymer Electrolytes for Dendrite-Free Zinc-Ion Batteries with High Performances and Prolonged Lifetimes

**DOI:** 10.3390/polym15010212

**Published:** 2022-12-31

**Authors:** Zhuoyuan Zheng, Haichuan Cao, Wenhui Shi, Chunling She, Xianlong Zhou, Lili Liu, Yusong Zhu

**Affiliations:** School of Energy Science and Engineering, Nanjing Tech University, Nanjing 211816, China

**Keywords:** hydrogel–polymer electrolyte, zinc-ion battery, zinc–alginate (ZA), dendrite-free

## Abstract

Aqueous zinc-ion batteries (ZIBs) represent an attractive choice for energy storage. However, ZIBs suffer from dendrite growth and an irreversible consumption of Zn metal, leading to capacity degradation and a low lifetime. In this work, a zinc–alginate (ZA) hydrogel–polymer electrolyte (HGPE) with a non-porous structure was prepared via the solution-casting method and ion displacement reaction. The resulting ZA-based HGPE exhibits a high ionic conductivity (1.24 mS cm^−1^ at room temperature), excellent mechanical properties (28 MPa), good thermal and electrochemical stability, and an outstanding zinc ion transference number (0.59). The ZA-based HGPE with dense structure is proven to benefit the prevention of the uneven distribution of ion current and facilitates the reduction of excessive interfacial resistance within the battery. In addition, it greatly promotes the uniform deposition of zinc ions on the electrode, thereby inhibiting the growth of zinc dendrites. The corresponding zinc symmetric battery with ZA-based HGPE can be cycled stably for 800 h at a current density of 1 mA cm^−2^, demonstrating the stable and reversible zinc plating/stripping behaviors on the electrode surfaces. Furthermore, the quasi-solid-state ZIB with zinc, ZA-based HGPE, and Ca_0.24_V_2_O_5_ (CVO) as the anode, electrolyte, and cathode materials, respectively, show a stable cyclic performance for 600 cycles at a large current density of 3 C (1 C = 400 mA g^−1^), in which the capacity retention rate is 88.7%. This research provides a new strategy for promoting the application of the aqueous ZIBs with high performance and environmental benignity.

## 1. Introduction

The demand for energy is constantly increasing with the rapid growth of the population. The production of energy from fossil fuels is no longer desirable and is gradually declining, not only because of the depleting global mineral reserves but also the corresponding serious environmental problems [1]. In recent decades, considerable efforts have been made to pivot energy supplies from traditional fossil fuels to renewable resources such as solar, wind, tidal, and geothermal energies [2]. Among them, rechargeable batteries, e.g., lithium-ion batteries (LIBs), occupy a unique and important position as energy storage systems [3], owing to their excellent energy density, good cycling performance, and high theoretical specific capacity [4].However, despite their extended applications, the flammable organic-liquid-electrolyte-based LIBs face numerous issues, e.g., environmental pollution, limited lithium resources, high cost, and potential safety problems, which limit their further development and motivate people to explore alternatives with high security, low cost, and environmental friendliness [5]. Aqueous rechargeable batteries are one of the most promising substitutes for grid-scale energy storage due to their high operational safety, environmental benignity, and high electrochemical performance [6,7]. Until now, various aqueous rechargeable batteries with different chemistries have been proposed, including alkali metal cations (e.g., Na^+^ and K^+^) [8] and multivalent charge carriers (e.g., Mg^2+^, Zn^2+^) [9].

With the continuous research and in-depth exploration of aqueous systems, aqueous zinc-ion batteries (ZIBs) have been identified as a promising choice [10]. The virtues of Zn anodes include low cost; high theoretical capacity (820 mAh g^−1^); mild, neutral-pH electrolyte; and feasible redox potential (−0.78 V vs. standard hydrogen electrode); etc. [11,12,13]. Previous studies regarding aqueous ZIBs have been primarily focused on their active materials, and fruitful achievements have been made, such as manganese- [14,15], vanadium- [16,17] and Prussian-based cathodes [18], as well as metallic zinc anodes [19,20].

As a pivotal component, the electrolyte also plays a vital role in battery chemistry since it is in charge of providing a basic operating environment, transporting ions, and connecting the two electrodes. However, traditional aqueous electrolytes confront plenty of challenges, such as notorious side reactions, liquid leakage, the continuous consumption of active zinc, and an irregular growth of zinc dendrites, which could cause severe safety problems such as a short circuit or cathode dissolution and corrosion, hindering their practical applications [21]. In recent years, increasing attention has been paid to electrolyte-improvement strategies [22]. Many effective approaches have been developed to eliminate the aforementioned issues, including the addition of functional additives [23], solvent optimization [24], and the development of gel/solid-state electrolytes [25], etc. Weina et al. introduced diethyl ether (Et_2_O) as an additive to improve the coulombic efficiency of a Zn–MnO_2_ battery [26], in which a high capacity-retention of 97.7% was obtained after 4000 cycles at 5 A/g; the highly-polarized Et_2_O molecules were found to suppress dendrite formation. Huayu et al. reported an acetamide-Zn (TFSI)_2_ eutectic electrolyte for ZIBs to assist in the formation of a stable, solid electrolyte interphase on a Zn anode [27]; the corresponding anode enabled reversible and dendrite-free zinc plating/stripping, even at high rate. Nevertheless, these approaches usually require complex fabrication processes and do not conform to the low-cost property of ZIBs.

A hydrogel–polymer electrolyte (HGPE) for ZIBs is a swollen polymer structure obtained by mixing zinc salt solutions with polymeric matrixes to achieve good Zn^2+^ diffusion as well as kinetic and high ionic conductivity [25]. With the restriction of water molecules, the water-induced side reactions such as hydrogen evolution can also be largely inhibited. In addition, HGPEs possess reasonable mechanical properties and can optimize the diffusion of Zn^2+^ so as to restrain the dendrite formation [28]. Qin et al. proposed a steric molecular combing strategy to synthesize a gel polymer electrolyte with a dynamic, self-adaptive interface [29]; theoretical simulations and experimental characterizations were conducted to reveal its strength in promoting the interfacial contact between the electrolyte and anode surface, as well as to determine the cycling lifespan of the battery. In our group’s previous study [30], a nylon-based composite gel membrane was fabricated via sequential, layer-by-layer electrospinning; the composite electrolyte had a high ionic conductivity, wide electrochemical window, and low activation energy, resulting in reversible ion dissolution/deposition behaviors.

Herein, a non-porous HGPE with zinc–alginate (ZA) as its matrix was prepared via the solution-casting method and ion displacement reaction. Alginic acid is a natural polysaccharide existing in the cytoderm of brown algae with characteristics such as an abundant yield, good biocompatibility, and easy degradation [31]. Additionally, it can interact with different metal ions (calcium, sodium, zinc, etc.) to form salt and absorb water molecules through plentiful hydroxyl groups, making it a potential material for use as an HGPE. The ZA-based polymer membrane shows a high uptake to the water electrolyte, high mechanical properties, good thermal stability, suitable ion conductivity, and a remarkable Zn^2+^ migration number. The ZA-based HGPE with a dense structure is proven to benefit the prevention of uneven ion migration and the reduction of excessive interfacial resistance. Moreover, it greatly promotes the uniform deposition of Zn^2+^ on the electrode surfaces, thereby inhibiting the growth of Zn dendrites. By adopting the prepared HGPE, the Zn symmetric battery shows a reversible zinc plating/stripping behavior (up to 800 h) at a current density of 1 mA cm^−2^ and a capacity of 1 mAh cm^−2^. The Zn dendrite suppression is further demonstrated in Zn/CVO cells with a superior cycling stability and a high reversible capacity. This research provides a new strategy for developing novel hydrogel–polymer electrolytes for aqueous ZIBs with good electrochemical performance, high security, and low cost.

## 2. Materials and Methods 

### 2.1. Materials

A.R. (analytical reagent)-grade solidum alginate, CaCl_2_, V_2_O_5_, poly tetra fluoroethylene (PTFE), and butyl alcohol (CH_3_(CH_2_)_3_OH) were obtained from Aladdin (Shanghai, China). A.R.-grade zinc sulfate heptahydrate (ZnSO_4_·7H_2_O) was supplied by Macklin (Shanghai, China). Acetic acid (CH_3_COOH), Zn metal foil (≥99.99%), and acetylene black were acquired from Sinopharm Chemical Reagent (Shanghai, China). The glass-fiber separator (GF/F) was purchased from Whatman (Shanghai, China). All chemical materials were used as received without further purification.

### 2.2. Preparation of ZA-Based HGPE

The ZA membrane was prepared using the solution-casting method. First, 0.4 g of solidum alginate powder was completely dissolved in 40 mL of ultrapure water and stirred at 500 rpm to form a homogenous solution. The homogenous solution was placed for 4h to remove the residual bubbles after stirring. Afterwards, the solution was poured into a glass Petri dish and dried on a hot plate at 50 °C for 6 h to evaporate the solvents. The achieved solidum alginate membrane was tailored to circular pieces (d = 19 mm). The obtained circular pieces were then immersed in a ZnSO_4_-based electrolyte (ZnSO_4_ aqueous solution, 3mol L^−1^) to conduct the displacement reaction. After standing overnight, the sample was rinsed with ultrapure water and then dried to finally obtain a ZA membrane with an average thickness of about 25 µm. The ZA membranes were saturated with a ZnSO_4_-based electrolyte for 24 h. Further measurement could be carried out with the obtained ZA-based HGPE.

### 2.3. Synthesis of Ca_0.24_V_2_O_5_·0.83H_2_O Nanobelt (CVO) Electrode

The detailed procedures are the same with the reported literature [32]. A mass of 356 mg of commercial V_2_O_5_ and 111 mg of commercial CaCl_2_ were first dissolved in 30 mL (1.17 M) of an aqueous solution of acetic acid to form a homogeneous solution. The solution was then transferred into a 50 mL Teflon-lined autoclave, which was put in a constant-temperature oven and maintained at 200 °C for 72 h. After cooling, the sample was collected and rinsed with ethanol and water. It was then dried in a vacuum at 50 °C for 6 h. Finally, the CVO electrode was constructed by placing the mixture of 70 wt% active material, 20 wt% acetylene black, and 10 wt% PTFE on the stainless-steel mesh and drying it at 60 °C for 12 h in the oven.

### 2.4. Physical Characterization

The following measurements were performed at room temperature unless otherwise stated.

The surface and cross-section morphology of the ZA polymer membrane and commercial glass-fiber separator were observed using a scanning electron microscope (SEM, Phenom ProX, Shenzhen, China) after being gold-sprayed. The samples were submerged in liquid nitrogen for the cross-section observation. The mechanical properties of the materials were tested by utilizing a Sansi UTM4304 (Shenzhen, China) electronic universal-testing machine at a speed of 5 mm min^−1^ [33]. Thermogravimetric analysis (TGA) and differential thermal analysis (DTA) of the membranes were carried out in a NETZSCH-STA409 instrument (Selb, Germany) to determine the thermal stability.

After the dried ZA membrane or glass-fiber separator were immersed in n-butanol for more than 12 h, the porosity (*P*) was calculated according to Equation (1):(1)$$P=\frac{m_{2}-m_{1}}{\rho V}$$
where *m*_1_ and *m*_2_ represent the weights of the ZA membrane and glass-fiber separator before and after absorption of n-butanol, respectively; *ρ* is the density of n-butanol (0.8098 g cm^−3^ at 25 °C), and *V* is the volume of the membrane or separator, which is calculated with the radius and the thickness of the samples.

The uptake to the ZnSO_4_ electrolyte (*η*) of the membranes was measured via Equation (2):(2)$$\eta =\frac{W_{t}-W_{0}}{W_{0}}\times 100\%$$
where *W*_0_ and *W_t_* represent the mass of the ZA membrane and glass-fiber separator before and after the absorption of the liquid electrolyte, respectively.

### 2.5. Electrochemical Measurement

The following measurements were performed with a CHI660C electrochemical working station (Chenhua, Shanghai, China) at room temperature unless otherwise indicated.

Using the blocking-type cells with stainless steels (SS) as electrodes, the electrochemical impedance spectroscopy (EIS) of the ZA-based HGPE or wet glass-fiber separator was examined at different temperatures (25–75 °C) in the frequency range of 1–100 kHz (step potential: 5 mV). Based on the results of EIS, the ionic conductivity of the electrolytes was calculated according to Equation (3):(3)$$\sigma =\frac{l}{R_{b}A}$$
where *σ* represents the ionic conductivity, *l* denotes the thickness of the ZA-based HGPE or glass-fiber separator, *R_b_* represents the bulk resistance from EIS, and *A* is the contact area of the stainless-steel electrode that is in contact with the ZA-based HGPE or glass-fiber separator.

Based on Evans’ technique [34], the Zn^2+^ ion migration number of the ZA-based HGPE or wet glass-fiber membrane was estimated. The *t*_*Zn*_^2+^ was calculated by Equation (4):(4)$$t_{Zn^{2+}}=\frac{I_{s}\left(\Delta V-I_{0}R_{0}\right)}{I_{0}\left(\Delta V-I_{s}R_{s}\right)}$$
where *I*_0_ and *I_s_* are the currents at the initial and steady states, respectively; *R*_0_ and *R_S_* represent the cell resistance before and after polarization; and ∆*V* is the step potential.

The electrochemical stability windows of the electrolytes were measured on the Zn/ZA-based HGPE or wet glass-fiber separator/SS cells by linear sweep voltammetry (LSV), which was carried out in the potential range of 0–2.1 V (vs. Zn^2+^/Zn) with a scan rate of 2 mV S^−1^.

A galvanostatic cycling test was performed with the symmetric cells (Zn/ZA-based HGPE or wet glass-fiber separator/Zn) to evaluate the Zn^2+^ plating/stripping stability between the Zn/electrolyte interface. The symmetrical cells were cycled at a constant current density of 1.0 mA cm^−2^ and an areal capacity of 1.0 mAh cm^−2^.

The electrochemical performance of the ZA-based HGPE was investigated using CR2025-type button cells (Kejing, Hefei, China) on a Land battery test system. The coin cells were assembled by sandwiching the ZA between the Zn metal anode and the CVO cathode. To study the cyclic behaviors, the cells were charged and discharged at a current density of 0.2 and 3 C (1 C = 400 mA g^−1^) constantly between 0.6 V and 1.6 V (vs. Zn^2+^/Zn). For the rate evaluation, the cells were run for five cycles under the current densities of 0.2 C, 0.5 C, 1 C, 2 C, 3 C, 5 C, and 10 C, and then back to 0.2 C.

## 3. Results and Discussion

The photographs of the prepared ZA membrane and the commercial glass-fiber separator are shown in Figure 1. The surface of the ZA membrane is transparent and dense (Figure 1a). As can be observed in Figure 1c, the surface of the ZA membrane is flat, and no obvious porous structure appears. This is distinctly different to the surface of glass-fiber separator, which has a large number of interlaced glass-fiber filaments (Figure 1e). The nonporous and compact structures of the ZA membrane are also proved by the porosity (0.03%) calculated via Equation (1). The nonporous surface of the ZA membrane can maintain a good interface contact with the Zn metal anode to assure uniform zinc-ion flux distribution; therefore, Zn dendrite formation and the pulverization of Zn metal can be restrained by the gel membrane. As the one of widely applied commercial separators, the glass-fiber separator owns uniform pore distribution to ensure even current densities in batteries. The porosity of the glass-fiber membrane is calculated by the porosity test to be 58.1%. By observing the cross-section of the ZA membrane (Figure 1d), it is found that its thickness is about 25 μm, which is much thinner than the glass fiber separator (Figure 1f, approximately 290 μm).

Solid-state electrolytes with excellent mechanical strength are confirmed to benefit stable Zn deposition and suppress metal dendrite growth [35,36]. Additionally, the GPEs or separators should have the appropriate tensile strength to meet the requirements of assembly and improve the security of batteries [37]. The mechanical properties of the ZA membrane and the glass-fiber separator are obtained through the tensile test and the results are shown in Figure 2a. The tensile strength and the breaking–elongation ratio of the ZA dry membrane are 28 MPa and 1.75%, respectively. Correspondingly, those of the dry glass-fiber separator are 1 MPa and 1.5%, respectively. The mechanical properties of the ZA membrane and the glass-fiber membrane after absorbing the ZnSO_4_-based electrolyte are shown in Figure 2b. The tensile strength and the breaking elongation ratio of the ZA-based HGPE are 13.5 MPa and 2.1%, respectively. On the contrary, the mechanical properties of the wet glass-fiber separator hardly changed after absorbing the electrolyte. Consequently, the obtained HGPE far exceeds the glass-fiber separator in terms of its mechanical strength, which has a better mechanical inhibition effect on the growth of zinc dendrites on the metal anode as well as fulfills the requirements of ZIB manufacturing and application.

The thermal stability of the separator is directly related to the operating temperature, the electrochemical performance, and the safety of ZIBs. The thermal stability of the ZA membrane and the glass-fiber separator was evaluated by thermogravimetric (TG) and differential thermal analyses (DTA) at a temperature-rising rate of 10 °C min^−1^ from 25 to 600 °C under N_2_. As is shown in Figure 3a, the first thermal decomposition of the ZA occurs between 50 °C and 200 °C, which corresponds to a 10% weight loss. The mass loss of the ZA is obvious beyond 200 °C, and there is a sharp exothermic peak at 200 °C in Figure 3b, indicating that the ZA material is gradually decomposing. When it reaches 600 °C, the remaining mass is 36.4%. As is shown in Figure 3c,d, the ZA-based HGPE and the glass-fiber separator immersed in ZnSO_4_ electrolyte (i.e., the wet glass fiber) are also tested through TG and DTA. It can be observed that the ZA-based HGPE starts to volatilize the solvent slowly at 60 °C. When the temperature reaches 160 °C, the solvent is completely volatilized, and the weight loss is 59%. On the other hand, the wet glass-fiber separator starts to volatilize the solvent from 35 °C, and the solvent is completely volatilized when the temperature gradually rises to 125 °C, with a remaining mass of 45%. The solvent evaporation process of the ZA-based HGPE is slower than that of the commercial glass-fiber separator, which indicates that it has a better electrolyte-retention capacity due to the existence of many hydrogen bonds as well as the improved thermal stability of the system.

Ionic conductivity is regarded as one of the most crucial parameters for evaluating the performance of electrolytes. The ionic conductivity of GPEs highly depends on the uptake of liquid electrolytes [38]. The high porosity (58.1%) of the glass-fiber separator ensures a high liquid absorption rate (179%). In contrast, the proposed ZA membrane is non-porous and compact; however, its liquid absorption rate can still reach up to 79% (Equation (2)). This is mainly because the large number of hydroxyl groups in the ZA molecule chain are capable of building hydrogen bonds with water molecules, forming the hydrogel–polymer electrolyte. The ionic conductivity of a ZA-based HGPE at room temperature is calculated to be 1.24 mS cm^−1^, and that of the liquid-electrolyte-glass-fiber separator system is 15.6 mS cm^−1^. The low ionic conductivity of the ZA-based HGPE can be attributed to two aspects: the ZA membrane has a relatively lower electrolyte uptake due to its non-porous, dense structure; in addition, the thickness of the ZA membrane is about 25 μm, approximately 1/12 that of glass-fiber separator, leading to a lower conductivity (Equation (3)).

The ionic conductivities of the ZA-based HGPE and wet glass-fiber separator at different temperatures from 25 °C to 75 °C are further measured, as is illustrated in Figure 4, in which the insets are the impedance plots of the two separators. The bulk resistances of the ZA-based HGPE and wet glass fiber can be acquired from the intercept of the straight line on the real axis of EIS shown in the insets of Figure 4 [39]. The ionic conductivities of the ZA-based HGPE and the wet glass-fiber separator increase with the temperature, which can be reasonably explained by the Arrhenius ion-conduction mechanism. According to the Arrhenius formula (*σ* = *A*exp(−*Ea*/*RT*)), the approximate linear relationship between Log*σ* and 1000/T for both separators can be achieved. The activation energy, Ea, of the ZA-based HGPE is 12.54 KJ mol^−1^, and that of the wet glass-fiber separator is 6.671 KJ mol^−1^, suggesting that the movement of Zn^2+^ ions in the HGPE requires more energy than in the liquid electrolyte. The slightly larger Ea is due to the comparatively lower electrolyte absorption of the ZA membrane, as well as its semi-solid state.

According to the space charge field theory, the contribution of Zn^2+^ cations to the overall ionic conductivity can be assessed by the ion transference number (t_Zn_^2+^); when t_Zn_^2+^ in the electrolyte is closer to one, the concentration polarization inside the battery becomes smaller, leading to an improved charge–discharge capacity and a reduced growth of zinc dendrites [40]. Chronoamperometry profiles of the zinc-symmetric cells with two different electrolytes are displayed in Figure 5 to measure t_Zn_^2+^, in which the insets illustrate the alternating impedance spectroscopy of the cells before and after the battery polarization. The t_Zn_^2+^ values of the ZA-based HGPE and wet glass-fiber separator are calculated to be 0.59 and 0.21, respectively, according to Equation 4. The high Zn^2+^ transference number in the ZA-based HGPE can be explained by the formation of a hydrogen bond between the hydroxyl groups of ZA and water molecules which greatly decreases the polarization effects in the electrolyte, enhances the interfacial compatibility, and reduces the interfacial impedance between the electrolyte and Zn anode. Therefore, even though it requires more activation energy, most of the current is used to drive the transportation of Zn^2+^ ions, leading to a larger t_Zn_^2+^ when compared to the wet glass-fiber separator. In other words, the proposed ZA-based HGPE is more preferable for the migration of Zn^2+^ cations than the SO_4_^2−^ anions and water molecules [41,42,43].

The electrochemical stability of the electrolyte determines the charge/discharge voltage window and energy density of the batteries. From the liner sweep voltammograms shown in Figure 6, no noticeable oxidative current is observed within the range of 0–1.5 V (vs. Zn^2+^/Zn) for both ZA-based HGPE and the wet GF separator due to be saturated by the same aqueous electrolyte, indicating that their electrochemical stability windows are similar (being 1.5 V, suitable for the applications in ZIBs). The water in the ZnSO_4_-based aqueous electrolyte begins to decompose to oxygen and hydrogen when the voltage is larger than 1.5 V (vs. Zn^2+^/Zn) and the oxidative current appears in the LSV.

The practical utilization of the Zn anodes in ZIBs has been hampered by the dendrite formation during cycling. The nonuniform Zn^2+^ plating/stripping between the electrolyte and Zn-anode surface is believed to be the dominating factor for Zn dendrite growth [44], which could eventually pierce through the separator, resulting in internal short circuits. To evaluate the reversibility of the Zn plating/stripping, the galvanostatic cycling test is conducted using the Zn/ZA-based HGPE or wet separator/Zn symmetric cells at a current density of 1 mA cm^−2^ and an areal capacity of 1.0 mAh cm^−2^. The obtained time-dependent voltage profiles of the symmetric cells are shown in Figure 7. The initial polarization voltage of the Zn/ZA-based HGPE/Zn symmetric battery is about 30 mV, and the cycle life is 800 h. During the deposition and stripping period, the voltage curve is relatively stable, and the polarization voltage is stable at about 15 mV at 800 h (Figure 7d), which fully shows that Zn^2+^ ions can be deposited/stripped uniformly and stably on zinc metal. On the contrary, the initial polarization voltage of Zn/wet glass-fiber separator/Zn symmetric battery is about 70 mV. During 450 h of charge and discharge, the polarization voltage experienced a process of decreasing and then increasing, mainly because of the poor adhesion between the zinc metal and the separator. This uneven deposition of zinc leads to the growth of zinc dendrites and the thickening of the SEI layer. At 443 h, the polarization voltage curve begins to show relatively large fluctuations (Figure 7c) and the polarization voltage becomes unstable, which is mainly due to the fact that the grown zinc dendrites pierce the separator, short-circuiting the battery. The outstanding dendrite suppression of the proposed ZA-based HGPE is further confirmed by the SEM morphology of the zinc electrodes after long cycles. The Zn/ZA-based HGPE or wet glass-fiber separator/Zn cells are disassembled after 400 h of cycling to investigate the surfaces of the metal zinc anodes, as is shown in Figure 7f,g. The dense and uniform SEI (solid electrolyte interface) with a relatively smooth and flat morphology can be observed in the surface of the zinc electrode of the Zn/ZA-based HGPE/Zn symmetric battery, whereas irregular and broken dendrites grow on the surface of zinc electrode in the Zn/wet glass fiber/Zn symmetric battery due to the uneven deposition and dissolution of Zn^2+^. These results prove that the ZA-based HGPE can achieve more stable and reversible zinc deposition because of its excellent mechanical properties, outstanding ion transference number and ionic conductivity, and good interfacial compatibility with metal anode, resulting in the inhibition of the growth of zinc dendrites.

For further assessing the performances of the ZA-based HGPE in ZIBs, the Zn/ZA-based HGPE /CVO and Zn/glass fiber saturated by ZnSO_4_ liquid electrolyte/CVO cells are assembled. The CVO materials are synthesized based on Section 2.3. Appendix A illustrate the XRD pattern and the SEM morphology of the as-prepared CVO cathode material, demonstrating its crystal structure and the typical nanobelt microstructure. Appendix A compares the cyclic voltammetry (CV) curves of the Zn/ZA-based HGPE/CVO and Zn/glass fiber saturated by the ZnSO_4_ liquid electrolyte/CVO cells. Multiple pairs of redox peaks are observed, proving the multistep reaction mechanism associated with Zn^2+^ insertion/extraction, so that the CVO can be used as a compatible, cathode-active material in the ZIB system. Appendix A exhibits the CV curves of the CVO electrode at different scan rates from 0.1 to 0.5 mV s^−1^. With increasing scan rates, the CV curves maintain a similar shape with slight shifting of redox peaks.

Figure 8 shows the rate performances (0.2 C, 0.5 C, 1 C, 2C, 3 C, 5 C, 10 C, and 0.2 C) of the batteries using ZA-based HGPE and wet glass fiber at room temperature (1 C = 400 mAh g^−1^). The discharge specific capacities of the two cells at different C-rates are separately 274.6 mAh g^−1^ vs. 259.4 mAh g^−1^ (0.2 C), 251.2 mAh g^−1^ vs. 246.8 mAh g^−1^ (0.5 C), 247.9 mAh g^−1^ vs. 227.3 mAh g^−1^ (1 C), 204 mAh g^−1^ vs. 189.1 mAh g^−1^ (2 C), 167.7 mAh g^−1^ vs. 145.7 mAh g^−1^ (3 C), 132.9 mAh g^−1^ vs. 105.2 mAh g^−1^ (5 C), and 40.2 mAh g^−1^ vs. 25.3 mAh g^−1^ (10 C). The discharge capacities of cells gradually decrease with the increased C-rates, and later recover to their original values when the currents return to 0.2 C. Moreover, the overpotential differences between the charging and discharging curves in the cell with ZA-based HGPE are obviously lower than those in the wet glass-fiber cell, especially at large C-rates. This is due to the excellent interfacial compatibility between the HGPE and the zinc anode. On the contrary, although the wet glass-fiber separator has a smaller internal resistance (Figure 4), this should be attributed to the movements of both the Zn^2+^ cations and the SO_4_^2−^ anions. However, the transportation of the SO_4_^2−^ anions will not benefit the rate performances of the ZIBs. As a result, the batteries using ZA-based HGPE could deliver better rate performances and have higher reversible capacities at various rates.

Finally, the cyclic properties of the ZIBs assembled with a ZA-based HGPE are investigated. As is shown in Figure 9a, the Zn/ZA-based HGPE/CVO cell retains 83.8% of the initial capacity after 150 cycles at a current density of 0.2 C, and the coulombic efficiency remains at approximately 100%. In contrast, the Zn/wet glass-fiber separator/CVO cell retains only 60% of its initial capacity after 145 cycles, and, due to overcharge, the coulombic efficiency drops to 45.64% at 142 cycles and is only 5.34% after 145 cycles. Similarly, as is shown in Figure 9b, the Zn/wet glass-fiber separator/CVO cell can only stably cycle for 370 cycles at a current density of 3 C, after which the capacity decays to 82.3% of the initial capacity and the coulombic efficiency drops sharply to only 30.7% at 379 cycles. However, the coulombic efficiency and capacity retention of the Zn/ZA-based HGPE /CVO cell are about 100% and 88.7%, respectively, after 600 cycles of charge and discharge at 3 C, demonstrating the high cycle reversibility of zinc ions in the cell. Figure 9c,d show the charge and discharge curves of the Zn/ZA-based HGPE/CVO and the Zn/wet glass-fiber separator/CVO cells at different cycles at 0.2 C and 3 C. These phenomena can be explained by the fact that, on one hand, the ZA-based HGPE has a denser structure and better compatibility with the metal anode, leading to the uniform deposition of zinc on the anode surface and the inhibition of zinc dendrites; the corresponding Zn metal surface after 300 cycles shows a relatively smooth and flat morphology, as is illustrated in Appendix A. On the other hand, the formation of irregular dendrites on the surface of the zinc and the formation of dead zinc in the ZIB assembled with the wet glass-fiber separator, as is shown in Appendix A, damages the separator, causing the short-circuit of the battery. It is worth noting that the capacities increase in the initial stage of the 3 C cycling process, mainly due to the activation process and gelation process of the positive electrode material. The activation process lasts from the first circle to 35 cycles, and the discharge specific capacity reaches the peak value of 173 mAh g^−1^. After this, the cycle curves tend to be stable.

## 4. Conclusions

In this paper, a zinc–alginate membrane is prepared by casting method, and a ZnSO_4_ aqueous electrolyte is used to plasticize hydrogel–polymer electrolytes with compact structure, extremely thin thickness, and high mechanical properties. The zinc–alginate hydrogel–polymer electrolyte exhibits high ionic conductivity (1.24 mS cm^−1^ at room temperature), high mechanical strength (28 MPa) and an excellent zinc-ion migration number (0.59). The zinc–alginate hydrogel with a dense structure is proven to benefit the prevention of the uneven distribution of ion current and the reduction of excessive interfacial resistance within the battery. Additionally, it greatly promotes the uniform deposition of zinc ions on the electrode, thereby inhibiting the growth of zinc dendrites. Consequently, when compared with wet glass-fiber separator, it has a better cycle performance and rate performance. Based on the above advantages, the zinc–alginate hydrogel–polymer electrolyte also exhibits good zinc-dendrite-inhibition ability during long cycles, and the zinc symmetric battery using the hydrogel electrolyte can be cycled stably for 800 h. This experimental work provides a simple method for the preparation of hydrogel–polymer electrolytes for zinc-ion batteries with good electrochemical performance, high security, and low cost.

## Figures and Tables

**Figure 1 polymers-15-00212-f001:**
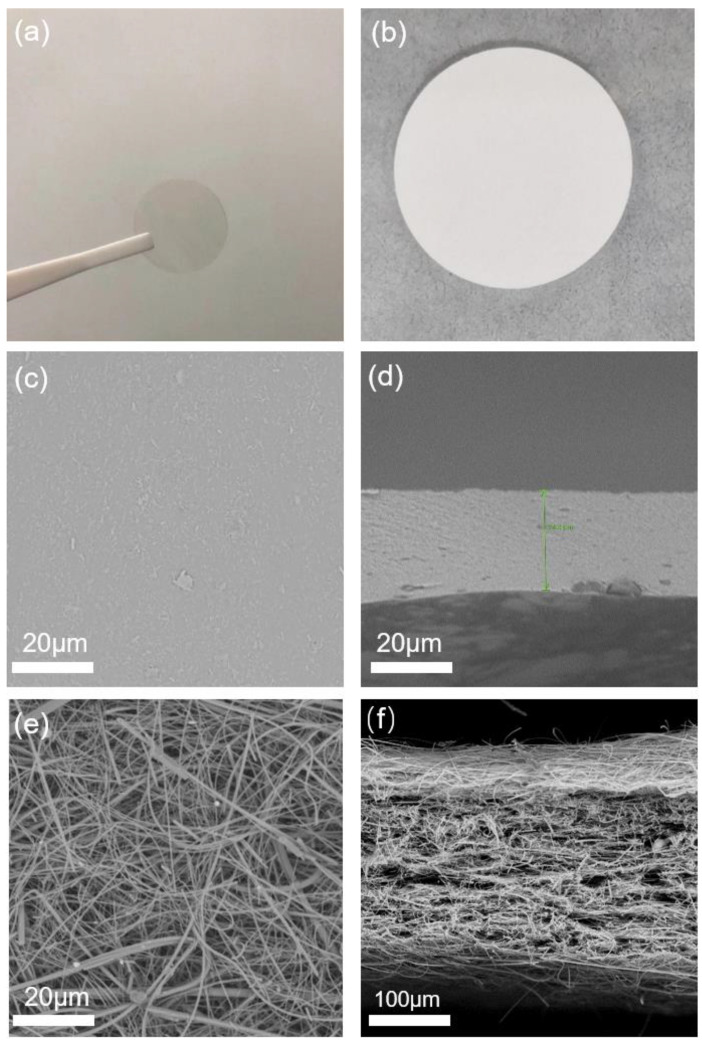
The photographs of (**a**) the prepared ZA membrane and (**b**) the glass-fiber separator. SEM images of (**c**) the surface of the ZA membrane, (**d**) the cross-section of the ZA membrane; (**e**) the surface of the glass-fiber separator, and (**f**) the cross-section of glass-fiber separator.

**Figure 2 polymers-15-00212-f002:**
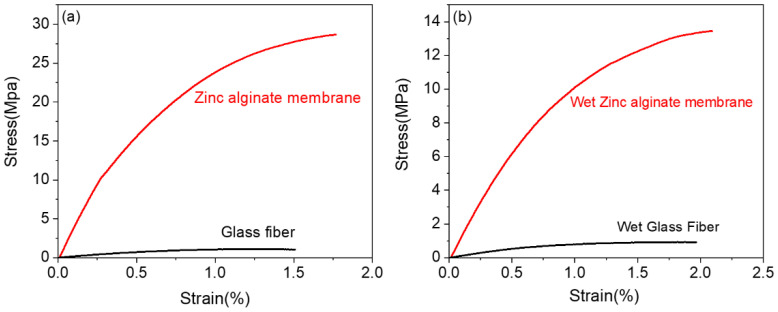
Stress–strain curves of the as-prepared ZA membrane and the glass-fiber separator (**a**) before and (**b**) after soaking in the ZnSO_4_ electrolyte.

**Figure 3 polymers-15-00212-f003:**
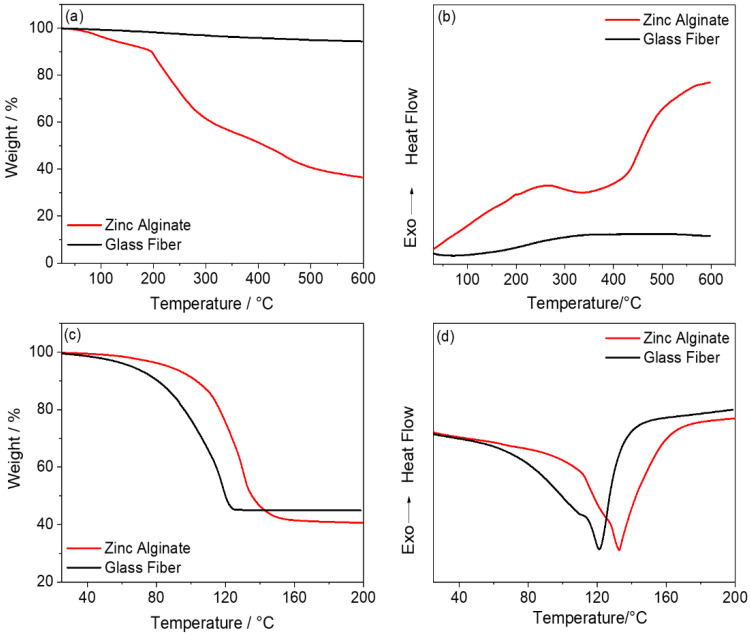
(**a**) Thermogravimetric (TG) and (**b**) differential thermal analysis (DTA) curves of the ZA membrane and the glass-fiber separator, (**c**) TG and (**d**) DTA curves of ZA membrane and the glass-fiber separator saturated with ZnSO_4_ electrolyte.

**Figure 4 polymers-15-00212-f004:**
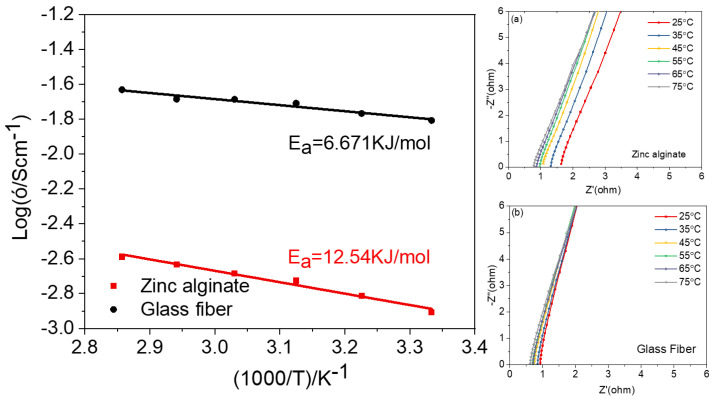
Arrhenius plot of the ZA-based HGPE and the glass-fiber separator saturated with ZnSO_4_ electrolyte. Insets are the impedance plots of the (**a**) ZA-based HGPE and (**b**) wet glass-fiber separator at different temperatures.

**Figure 5 polymers-15-00212-f005:**
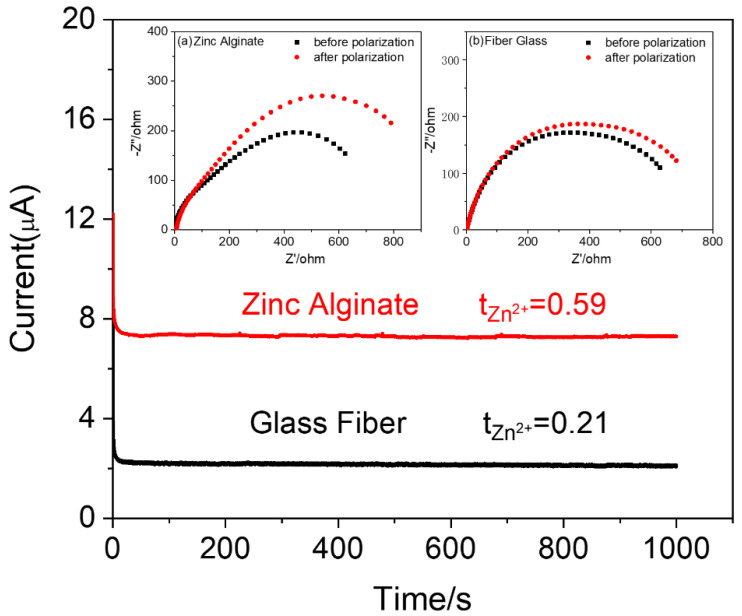
Chronoamperometry profiles for Zn/ZA-based HGPE/Zn and Zn/wet glass-fiber separator/Zn cells. Insets are impendence plots of (**a**) Zn/ZA-based HGPE/Zn and (**b**) Zn/wet glass-fiber separator Zn cells before and after polarization.

**Figure 6 polymers-15-00212-f006:**
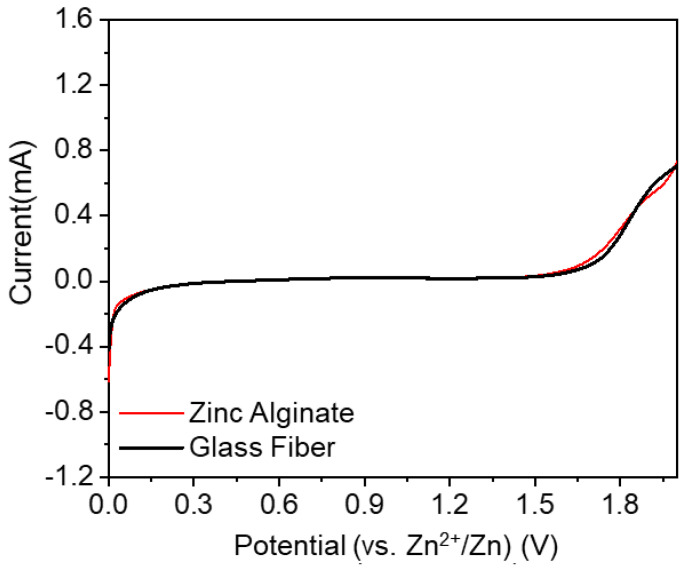
The linear sweep voltammograms of the ZA-based HGPE and the wet glass-fiber separator.

**Figure 7 polymers-15-00212-f007:**
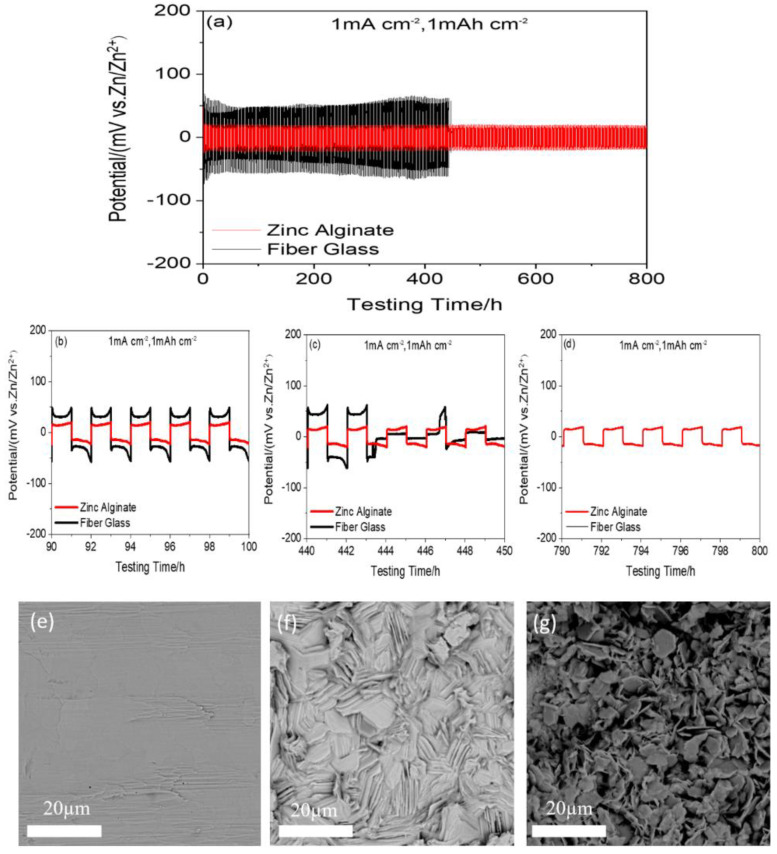
Electrochemical performance of Zn/ZA-based HGPE/Zn and Zn/wet glass-fiber separator/Zn symmetrical cells: (**a**) Zinc-ion plating/stripping voltage profiles of two symmetric cells at a current density of 1 mA cm^−2^ and a capacity of 1 mAh cm^−2^. Detailed plating/stripping voltage profiles at the (**b**) 90th–100th, (**c**) 440th–450th cycles, and (**d**) 790th–800th cycles. The SEM image of (**e**) the pristine Zn electrode and the Zn metal surfaces after 400 cycles in (**f**) Zn/ZA-based HGPE/Zn and (**g**) Zn/wet glass fiber separator/Zn cells.

**Figure 8 polymers-15-00212-f008:**
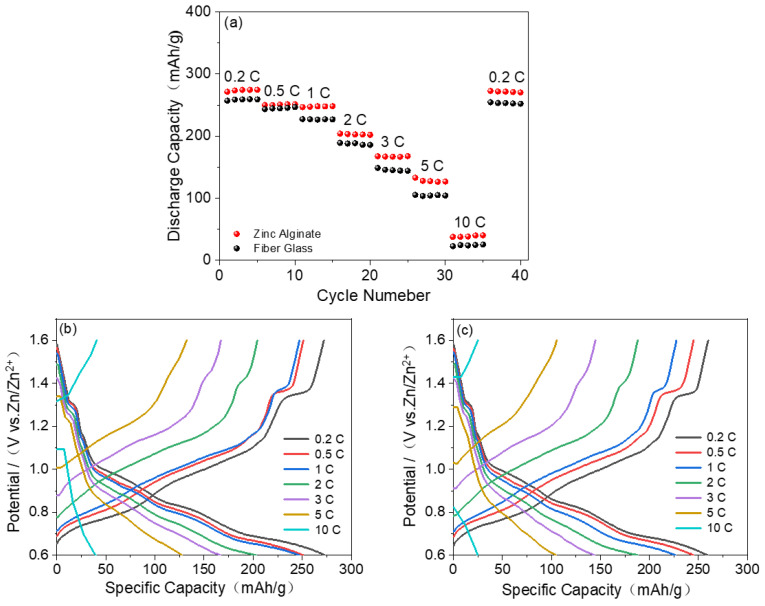
Electrochemical performances of the Zn/ZA-based HGPE/CVO and the Zn/wet glass-fiber separator/CVO at room temperature: (**a**) rate performances; charge–discharge curves of the Zn/CVO cells using (**b**) the ZA-based HGPE and (**c**) the wet glass-fiber separator.

**Figure 9 polymers-15-00212-f009:**
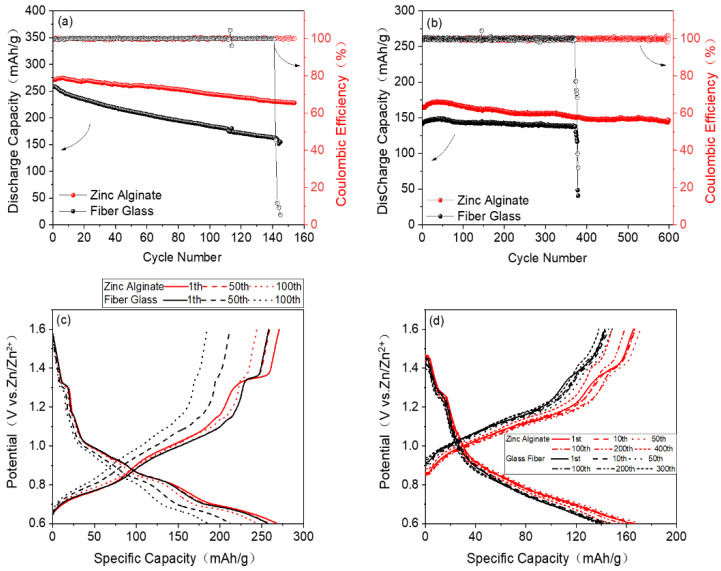
Electrochemical performance of the Zn/ZA-based HGPE/CVO and Zn/wet glass-fiber separator/CVO cells at room temperature: (**a**) 0.2 C and (**b**) 3 C. Charge and discharge curves of the Zn/ZA-based HGPE/CVO and Zn/wet glass-fiber separator/CVO cells at different cycles at (**c**) 0.2 C and (**d**) 3 C.

## Data Availability

The data presented in this study are available on request from the corresponding author.

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
