# Peer review of "Low-Cost Zinc–Alginate-Based Hydrogel–Polymer Electrolytes for Dendrite-Free Zinc-Ion Batteries with High Performances and Prolonged Lifetimes"

_polymers, 2022, doi:10.3390/polym15010212_

Round 1
Reviewer 1 Report
This research exhibited a Zn alginate-based Hydrogel Polymer Electrolyte to realize the dendrite-free Zn metal anode and then the improved performance of full CVO//Zn batteries. In specific, the physicochemical properties are characterized in detail by the alginate-based hydrogel polymer electrolyte, especially in terms of morphology, ionic conductivity, mechanical properties, etc. Regarding the electrochemical performance, Zn metal anodes also show good performance as well as the elongated full cell performance. In my opinion, this report is comprehensive to provide new ideas about polymer alginate-based materials for zinc ion batteries. So I support the publication of this paper. Before I 'm completely positive, the following issues should be addressed:
1. As for Figure 4, the activation energy of polymer alginate electrolyte is larger than that in glass fiber, that is 12.54 vs 6.671 KJ mol-1, indicating that zinc ions are more difficult to transfer in Alginate electrolyte. However, the transference number of zinc ions in polymer alginate electrolyte is larger than in glass fiber (Figure 5). These two data seem to be contradictory. The author needs to give a detailed explanation
2 Regarding the LSV results (Figure 6), the oxidative current and the reductive current should be explained, especially for the relatively narrowed window of the electrolyte.
3. For EIS resistance (Figure 4), the internal resistance of alginate-based electrolyte is larger than that of glass fiber. However, in terms of rate performance (Figure 8 b and 8c), the polarization of glass fiber is larger compared to the alginate-based electrolyte. So, how can the larger internal resistance be compromised when conducting the rate performance?
Reviewer 2 Report
The paper entitled " Low-cost Zinc Alginate-based Hydrogel Polymer Electrolyte for Dendrite-free Zinc-ion Batteries with High Performances and Prolonged Lifetime" has a very interesting results. The authors provided a new strategy to develop novel hydrogel polymer electrolytes for aqueous ZIBs with good electrochemical performance, high security, and low cost. The Abstract and the introduction are clear and sufficient. The Experimental technique is written in a good manner. My opinion, the paper is accepted to be published in present form.
Author Response
Thank you very much for your kind comment.
Reviewer 3 Report
The paper is written well but presentation is poor in the sense that from literature its lacking novelty.
Similar papers are already in literature where most of studies looking similar. Likewise
Self-healable hydrogel electrolyte for dendrite-free and self-healable zinc-based aqueous batteries - ScienceDirect
Batteries | Free Full-Text | SiO2-Alginate-Based Gel Polymer Electrolytes for Zinc-Ion Batteries (mdpi.com)
Novelty point of view its difficulty to judge however some merit of papers could put this paper inline. My specific comments are
1. FTIR must incorporated to understand the mechanism of interaction
2.Impedance spectra i.e. conductivity mechanism must be in detail like temperature dependent conductivity, conductivity vs composition plot is necessary to understand the mechanism
3.Surface morphology is needed to understand the electrode/electrolyte interface
Round 2
Reviewer 1 Report
In the revised report, the authors claimed the movement of SO42- anions can be hindered in the ZA-HGPE, while the Zn2+ cations can move more freely. This claim might resolve my questions (Q1 and Q3). It seems reasonable, however, no experiments or theoretical results or supportive reports are provided. So, I am confused here about how the authors can make such a claim.
Reviewer 3 Report
Revised paper is accepted
Author Response
Thank you very much for the kind comment!